# Design, Engineering and Discovery of Novel α-Helical and β-Boomerang Antimicrobial Peptides against Drug Resistant Bacteria

**DOI:** 10.3390/ijms21165773

**Published:** 2020-08-11

**Authors:** Surajit Bhattacharjya, Suzana K. Straus

**Affiliations:** 1School of Biological Sciences, 60 Nanyang Drive, Nanyang Technological University, Singapore 637551, Singapore; 2Department of Chemistry, University of British Columbia, 2036 Main Mall, Vancouver, BC V6T 1Z1, Canada

**Keywords:** multi-drug resistant (MDR) bacteria, extensively drug resistant (XDR) bacteria, antimicrobial peptides (AMPs), lipopolysaccharide (LPS) binding, α-helical and β-boomerang AMPs

## Abstract

In an era where the pipeline of new antibiotic development is drying up, the continuous rise of multi-drug resistant (MDR) and extensively drug resistant (XDR) bacteria are genuine threats to human health. Although antimicrobial peptides (AMPs) may serve as promising leads against drug resistant bacteria, only a few AMPs are in advanced clinical trials. The limitations of AMPs, namely their low in vivo activity, toxicity, and poor bioavailability, need to be addressed. Here, we review engineering of frog derived short α-helical AMPs (aurein, temporins) and lipopolysaccharide (LPS) binding designed β-boomerang AMPs for further development. The discovery of novel cell selective AMPs from the human proprotein convertase furin is also discussed.

## 1. Introduction

Antimicrobial peptides (AMPs) protect organisms from all kingdoms of life against infections. The innate immune system of humans and other mammals serves as a first line of defense against a wide range of invading pathogens that may encompass bacteria, fungi, viruses, and parasites [1,2,3,4]. AMPs are an integral component of the host defense innate immunity. The defensive role of AMPs against infections in humans has been underscored by the observations that down regulation of the production of AMPs e.g., LL37, defensins, yielded increased susceptibility of bacterial diseases [5,6,7]. Apart from their potential direct in vivo killing of pathogens, multiple other biological functions are linked to AMPs, which are consequently also termed Host Defense Peptides (HDPs) [3]. AMPs can display a range of immunomodulatory activities [2,8,9], including anti-inflammatory, pro-inflammatory, adaptive immunity, chemotaxis, and endotoxin neutralization. AMPs can mediate immunomodulation, and have, although indirectly, been connected to the elimination of invading pathogens. Some AMPs are known to promote wound-healing, angiogenesis, and also to exert lethal effects against cancer cells [3,10,11,12]. Increasingly, the antibiofilm activity of AMPs has been characterized [13,14,15]. AMPs have been shown to be active in the inhibition, dispersal, and eradication of bacterial biofilms. The synergistic property of AMPs with conventional antibiotics in killing both planktonic and biofilm bacteria has been demonstrated [16].

Given the diverse activity of AMPs, research in the last few decades has focused on characterizing AMPs and finding active sequences, with the major goal of developing AMPs as peptide-based alternatives to antibiotics [17,18]. The surge of drug resistant, multidrug resistant (MDR), and extremely drug resistant (XDR) bacteria are a significant challenge to human health [19]. Novel antibiotics are urgently needed to treat infections that are caused by the ESKAPE (*Enterococcus faecium*, *Staphylococcus aureus*, *Klebsiella pneumoniae*, *Acinetobacter baumannii*, *Pseudomonas aeruginosa*, *Enterobacter cloacae*) group of pathogens. In 2019, the US centers for disease control and prevention (CDC) reported an estimated annual number of antibiotic-resistant infections of 2.8 million, resulting in 35,000 deaths [20]. Further, the CDC has highlighted carbapenem resistant Acinetobacter, Enterobacteriaceae, drug resistant *N. gonorrhoeae* as urgent threats. Drug resistant campylobacter, extended spectrum β-lactamase (ESBL) producing Enterobacteriaceae, multi-drug resistant *P. aeruginosa,* drug resistant Salmonella, vancomycin resistant Enterococcus (VRE) and methicillin resistant *S. aureus* (MRSA) belong to the serious threats category. The potential treatment options against drug resistant Gram negative bacteria are generally even more challenging than for Gram positive, due to the impermeable outer membrane LPS barrier found in Gram negative bacteria. Given the lack of available antibiotics, infections that are caused by extremely drug resistant (XDR) *P. aeruginosa* and *K. pneumoniae* must be treated with polymyxins, which are nephrotoxic and neurotoxic [21], with new delivery systems being developed to mitigate these unwanted effects [22].

The ability of AMPs to kill drug resistant bacteria and the relatively low occurrence of resistance development suggest that AMPs or AMP-based peptides have high potential for the treatment of antibiotic resistant bacteria. For example, Gram negative bacteria that are colistin resistant can be sensitive to AMPs: LL-37, cecropin A, and magainin 1 have demonstrated lethality against colistin resistant *K. pneumoniae* [23]. Gaining a mechanistic understanding of the plethora of activities of AMPs would be vital to fully comprehend structure-activity correlations. The mode of anti-bacterial activity of AMPs differs vastly from that of antibiotics which chiefly target intracellular macromolecules [24,25]. In many cases, both outer-LPS (in Gram negative) and inner bacterial membranes are permeabilized or disrupted upon binding to AMPs, leading to cell lysis. Indeed, a large proportion of AMPs have been shown to kill bacteria through this membranolytic process [1,4,26,27,28]. However, some AMPs can also target intra-cellular DNA or protein components, followed by a non-lytic cell killing process [29,30,31]. The acquisition of amphipathic secondary structure has been considered as the driving force of AMP bacterial plasma membrane binding. A net positive charge, under physiological conditions, establishes ionic interactions with the negatively charged bacterial membrane, whereas the hydrophobic residues facilitate insertion into the non-polar milieu of the membrane.

Using inner membrane mimics, several models e.g., barrel stave, toroidal pore, carpet/detergent-like were proposed describing how AMPs can disrupt integrity of bacterial plasma membrane [4,26,27,28]. The broad-spectrum activity and high effectiveness in bacterial cell killing by AMPs would also require potential destabilization of the outer components of bacterial membranes [32,33]. However, the outer peptidoglycan layer of Gram positive bacteria is less effective in preventing drug permeation. Indeed, antibiotics and AMPs can penetrate across peptidoglycan layers [34,35]. In contrast, the outer membrane of Gram negative bacteria serves as an efficient permeability barrier for antibiotics, host defence proteins, and peptides [36,37,38,39,40]. Outer membrane permeability is largely maintained by the highly organized structure of LPS in the outer leaflet [35,41,42]. The broad-spectrum activity of AMPs requires the permeabilization of both the outer and inner membrane of Gram negative bacteria. Atomic-resolution structures of potent AMPs in LPS micelles have revealed plausible disruption mechanisms of the outer membrane of Gram negative bacteria [43,44,45,46,47,48,49]. These investigations showed compact tertiary structures of AMPs and potential ionic and hydrophobic interactions with the lipid A domain of LPS [40,50,51,52,53].

A large number of studies have demonstrated that AMPs could be promising molecules for the treatment of drug resistant bacteria. However, to-date, very few AMPs are in the advanced stages of clinical trials [17,18,25] and most are being developed by smaller pharmaceutical companies. Reasons for these limited applications of AMPs are linked to the high cost of synthesis, low in vivo stability, host toxicity, and limited bioavailability. Consequently, strategies need to be employed to overcome these limitations in order to further develop the next generation of AMPs. The cost of chemical synthesis of AMPs can be minimized by shortening the length of AMPs through engineering and designing [54]. Large scale recombinant production of AMPs is also a viable option that would significantly reduce the cost of peptides [55]. Proteolytic degradation is likely to be the major cause of the short in vivo half-life of AMPs. Several strategies have been exploited for the generation of AMPs that are stable against proteases: e.g., cyclization of AMPs [56] and the introduction of disulfide bonds and covalent bonds between sidechains [57,58,59]. Alternative approaches to minimize proteolytic cleavage include producing synthetic AMPs that incorporate D-amino acids [60], β-amino acids [61], and other non-natural amino acids [62]. The topic of improving in vivo stability has been extensively reviewed elsewhere [4,63,64], and it will not be discussed further here. In terms of toxicity, some studies have shown that the introduction of D-amino acids or the rational optimization of amphipathicity can help in lowering host toxicity [60,65,66,67]. Furthermore, toxic AMPs were converted to non-toxic peptides by replacing hydrophobic residues with polar or Ala in the potential heptad repeat units [68,69]. Finally, the last shortcoming of AMPs that involves improving their bioavailability has only been addressed in a limited manner. Some studies have shown that enriching AMPs with Arg/Trp residues results in peptides with increased bioavailability [70]. Alternatively, the conjugation of AMPs with different polymers [71,72] or encapsulation in pegylated micelles [60] has been proposed as a feasible option to increase bioavailability [64].

Creating designed broad spectrum AMPs with high therapeutic index and improved bioavailability are of significance toward the development of peptide antibiotics. In the next sections, we describe engineering short α-helical AMPs, namely temporins and aurein, to yield an improved activity spectrum. We further discuss de novo designed β-boomerang AMPs that target the LPS outer membrane of Gram negative bacteria. Finally, human proteins or the proteome could be rich source of future AMPs. Hence, we present the discovery of non-hemolytic non-cytotoxic broad spectrum AMPs from the human protease furin.

## 2. Engineering Temporins for Superior Activity

Skin secretions of frogs are a rich source of a variety of AMPs, which differ in composition, length, structure, and activity profile [73,74,75,76]. Indeed, a considerable fraction of AMPs (i.e., ~1019 sequences) in the Antimicrobial Peptide Database (APD) are from frog [77]. Temporins are a group of short AMPs, 10 to 14 amino acids, which were first identified from screening the c-DNA library from the skin of the European red frog *Rana temporaria* [78,79,80]. Currently, there are 130 peptides listed under the family of temporins, some of which are isolated from different species of frog [81,82]. The antimicrobial activity and conformations of all the members of the temporin family have not yet been characterized. A peculiar feature of temporins that makes them strikingly different from many other AMPs is their low cationicity and high content of non-polar residues. Notably, the primary structures of some of the peptides are devoid of any basic amino acid residues [81,82]. To date, temporin-1Ta (TA), temporin-1Tb (TB), and temporin-1Tl (TL) have been the most widely investigated for activity, structure, synergy, rational design, engineering and SAR studies [83,84,85,86,87,88,89,90,91,92]. Table 1 shows amino acid sequences of temporins, as well as the designed analogs discussed here.

TA and TB were found to be active against Gram positive bacteria, including clinical isolates, while being only weakly hemolytic [79]. In addition, TA and TB also displayed growth inhibition of viruses, protoza, and parasitic activity in in vitro assays [93,94,95,96]. In vitro wound healing activity of TA and TB has also been reported [97,98]. However, TA and TB are unbale to exert any lethal effect on Gram negative bacteria. In contrast, TL was observed to be active against both Gram positive and Gram negative bacteria [80]. TL also displayed antifungal activity and it was toxic to certain cancer cells [99,100]. Unfortunately, TL has a low therapeutic index due to its high hemolytic and cytotoxic activities [84]. The multi-functional nature and short sequences of TA, TB, and TL have made these peptides attractive templates for the development of antimicrobial therapeutics. Extensive mechanistic [88,101,102,103,104] and structural studies [90,105,106,107] of TA, TB, and TL (Figure 1) provided essential information in the design of analogs.

TB and TA gained more attention for the development of broad spectrum AMPs due to their low hemolytic activity. The analogs of TB were chiefly designed to yield broad spectrum activity by increasing cationicity and hydrophobicity. An analog of TB was developed, termed TB_KKG6A (Table 1), through Ala scanning replacement and the introduction of basic Lys residues at the N-terminus. TB_KKG6A showed activity against Gram negative strain *S. typhimurium* (minimum inhibitory concentration (MIC) ~6 μM) and *E. coli* (MIC ~3 μM) [105]. In LPS micelles, TB_KKG6A did not show any aggregation and formed a monomeric helix-kink-helix structure [105,108]. TB_L1FK (Table 1) is a computationally designed analog of TB that contains a replacement of Leu1 to Phe, the deletion of residue Asn at position 7, and an addition of Lys13 at the C-terminus. Studies comparing the antibacterial activity of TB_KKG6A and TB_L1KF [109,110] showed that both analogs showed higher MIC values of 32 to 64 μM, against most of the Gram negative strains tested; however, TB_L1KF was found to have lower MIC values of 8 to 16 μM against strains of *Acinetobacter baumanii*. Another analog of TB, TB-YK (Table 1), was examined in vivo with TA peptide [111]. TB-YK displayed moderate Gram negative activity and had synergetic in vivo activity with TA in a murine infection model. Extensive positional Ala scanning of TA yielded 13 analogs, which were characterized in detail in order to determine whether relationships between antibacterial activity, hemolytic activity, conformations, hydrophobicity, cationicity, and hydrophobic moment could be found [112]. Overall, most analogs were active against Gram positive bacteria, with some analogs being more potent than the parent peptide. However, the most active analogs were also more hemolytic and characterized by more helical content. Analogs of TL were designed by modulating helicity, hydrophobicity, charge, and amphipathicity: those that contained multiple Pro or D-Pro residues at positions of Gln3 and Gly10 (Table 1) showed sufficiently reduced hemolytic activity and also lower helicity [113]. However, the antibacterial activity of these Pro analogs were appreciably reduced when compared to the native TL. In further investigations, TL analog 9 and 10 (Table 1) were designed to be non-hemolytic, but displayed poor activity against clinical isolates and drug resistant bacteria [113]. Substitution of residue Phe to Leu at positions 5 and 8 yielded an analog that retained much of the antibacterial activity with lower cytotoxicity [114].

As the LPS outer membrane can inactivate TA and TB, as well as other AMPs, a different strategy was developed to design peptides with broad spectrum activity. A six-residue aromatic/cationic peptide, WKRKRF, termed β-boomerang motif, because, upon interaction with LPS, it folds into a compact ‘boomerang’ structure, was incorporated into the temporins [115]. It was hypothesized that the inclusion of this motif into AMPs might be effective in abolishing LPS outer membrane induced aggregations, consequently resulting in broad spectrum activity. The LPS binding β-boomerang motif was added to the C-terminus of TA to give FG21 (Table 1), of TB to give LG21 (Table 1), and of another helical peptide to give KG20. As expected, all three peptides demonstrated broad spectrum activity against Gram negative and Gram positive bacteria [116]. It is noteworthy that FG21 and LG21 retained much of their antibacterial activity in LB medium containing 150 mM sodium chloride. In hemolysis assays, all three AMPs demonstrated limited lysis of human red blood cells (RBCs). Even at a concentration of 100 μM, the extent of hemolysis was found to be 8.5%, 9.9% and 2.4% for FG21, LG21 and KG20, respectively [116]. Furthermore, all three designed AMPs were bestowed with endotoxin neutralization potentially due to efficient interactions with LPS. Notably, LG21 demonstrated superior endotoxin neutralization activity in comparison to other two AMPs. An Ala substitution study further showed that residues Trp15, Arg19, and Phe20 were key to the broad spectrum antibacterial activity of the designed AMPs, which can be correlated with the absence of self-assembly in LPS micelles for the active analogues [116]. Furthermore, a separate study showed that individual replacement of other cationic residues namely Lys16, Arg17, Lys18 with Ala in the β-boomerang motif also yielded largely inactive analog peptides [117]. In contrast, Ala replacement of residues Asn7, Lys10 and Ser11 belonging to the TB part of LG21 did not cause any significant change in antibacterial activity. Therefore, these studies demonstrated that the β-boomerang motif is critical in determining broad spectrum antibacterial activity of the designed AMPs. The mechanism by which these AMPs kill bacterial cells has been shown to be the permeabilization of the outer membrane and inner membrane of Gram negative bacteria. Bacterial membrane permeabilization and dye entrapped liposome leakage assays performed on LG21 and the inactive versions LG21K16A, LG21R17A, LG21K18A, and LG21R19A revealed that LG21 displayed the largest effect toward bacterial membrane permeability followed by the other analogues in the following order: LG21 > LG21K16A > LG21R17A > LG21K18A > LG21R19A [117]. Similar observations were made in dye leakage assays using liposomes composed of 1-palmitoyl-2-oleoyl-phosphatidylcholine/phosphatidylglycerol (POPC:POPG) (3:1), which mimics the plasma membrane of bacteria. The atomic resolution structure of LG21 was determined in perdeuterated zwitterionic DPC micelles by NMR. In free solution LG21 assumes random conformations, whilst the micelle-bound state of LG21 is characterized by a well folded α-helical structure encompassing residues Ile4 to Phe20 [117]. The three-dimensional (3-D) structure of LG21 revealed that the β-boomerang motif also folded in a helical conformation and sustained packing interactions with the helical region of the TB part (Figure 2). The topology of the helical structure of LG21 resembles a ‘lollypop’ or a ‘drumstick’: the bulky head of the lollypop is constituted by the β-boomerang motif, whereas the tail part of the ‘lollypop’ is maintained by the TB segment. The helical structure of LG21 displays a large hydrophobic surface that is mainly sustained by non-polar residues of TB and residue W15 of the β-boomerang motif. The polar surface of the helix is rather short, occupied by the sidechain of residues Asn7, Lys10, and Ser11. The β-boomerang segment of the LG21 helix contains potential cation-π interactions in which the sidechains of the basic residues Arg19 and Lys16 interact with the aromatic sidechains of residues Trp14 and Phe20, respectively (Figure 2). Paramagnetic Relaxation Enhancement (PRE) NMR, using spin labelled lipid, indicated that the N-terminus of LG21 helix could be inserted into the hydrophobic core of micelles, whereas most of the residues in β-boomerang may reside at the micelle-water interface.

## 3. Engineering Aurein 2.2 for Superior Activity and Bioavailability

Another rich source of frog skin AMPs is the aurein peptide family. Aurein peptides are cationic antimicrobial peptides that are secreted from the Australian southern bell frogs *Litoria aurea* and *Litoria raniformis* [118]. They are comprised of five families of aurein peptides, which include the short and active aurein peptides from family 1 to 3, as well as longer and typically inactive family 4 and 5 peptides [118,119]. Aureins 1–3 exhibit moderate broad spectrum activity against pathogens, with higher activity against Gram-positive bacteria versus Gram-negative bacteria [118]. In addition, aurein 1–3 peptides have anticancer activity. Indeed, some peptides, such as aurein 1.2, 3.2, and 3.3, display their strongest activity against 30–50 different types of cancer [118]. Typically, aurein peptides range from 13 to 25 amino acid residues in length [120]. Specifically, aurein 2.2 and aurein 2.3 (Table 2) have a net charge of +2 and require an amidated C-terminus for activity [121]. They are active against Gram positive bacteria, such as *S. aureus* and *S. epidermidis* [121], with a MIC of 16–32 μg/mL (or 9–18 μM) against both. The N-terminus is required for activity, whereas truncation of three residues from the C-terminus has no effect on its antimicrobial function [122,123]. The truncation of the N-terminus makes aurein 2.2 immunomodulatory. Interestingly, a recent study comparing the structure and function of temporin L and aurein 2.5 [124] found that, although both AMPs adopt α-helical structure, they disrupt membranes differently. The results presented for aurein 2.5 [124] mirror earlier studies [121,123], which showed that, as long as the residue at position 13 in aurein 2.2 is hydrophobic (i.e., L, I, V, A or F), the resulting peptides have similar structures and modes of action (i.e., as aurein 2.2, 2.3, and 2.5 do).

The mechanism of action and structure of aurein 2.2 and aurein 2.3 have been extensively studied in recent years. Early studies showed the importance of studying bilayer perturbation in membrane models containing PG lipids, indicating that, as for most AMPs, electrostatic interactions are important in the lipid–peptide interaction [121]. Both aurein 2.2 and 2.3 adopt a highly α-helical structure, resulting in a hydrophobic length of ~24 Å. Given that a POPC:POPG bilayer has a thickness of ~39 Å, these AMPs were found to disrupt POPC:POPG lipid membranes via toroidal pore formation, a very plausible mechanism given the mismatch [120]. In contrast, in a DMPC/DMPG (1:1) membrane model, the peptides work in a detergent like model [120,122]. This further highlights the importance of the hydrophobic thickness of the lipid bilayer and the membrane composition for the mechanism of action. These mechanisms of action are different from aurein 1.2, which functions using a carpet mechanism [125,126]. In a more recent study, Wenzel at al. [127] showed that aurein 2.2 forms ion selective pores, permitting the translocation of ions, such as potassium and magnesium. In addition, aurein 2.2 also causes membrane permeabilization, which disrupts the membrane potential and decreases the energy supply of the cells leading to cell death. Wenzel et al. also confirmed that truncation of aurein 2.2 by three amino acids at the C-terminus (denoted henceforth as aurein 2.2-Δ3, Table 2) had no impact on the mechanism of action when compared to the full-length natural peptide [127].

Arg and Trp residues were used to systematically replace residues in the parent peptide in order to design more active versions of aurein 2.2-Δ3. Previous work had shown that cationic residues such as Arg and Lys mediate the initial electrostatic interaction between HDPs and the bacterial cytoplasmic membrane [128,129], thereby resulting in more active AMPs. Trp residues preferably bind in the interfacial region of lipid membranes, thereby facilitating peptide-lipid interactions [128]. In addition, the use of both Arg and Trp amino acids allows for the formation of cation-π interactions, which also serve to improve peptide-membrane interactions [129]. Finally, as mentioned earlier, enriching AMPs with Arg/Trp residues potentially results in peptides with increased bioavailability [70].

The ninety novel analogues were designed, such that the hydrophobic and basic amino acid residues of aurein 2.2-Δ3 were substituted with Trp and Arg, respectively, in order to increase favorable interactions. For most peptides, residues Asp4 and Ile5 were kept unchanged and generally, aromatic (e.g., Phe3) and/or hydrophobic residues (e.g., Val9) were replaced by Trp and lysines (i.e., Lys7 and Lys8) were replaced by Arg (Table 2). Residue Asp4 was preserved in most analogues, as it has been previously suggested that it may play an important role in ion selectivity of aurein 2.2 [127]. Single to multiple (max. five individual Trp; three individual Arg; or a mixture of eight Trp and Arg residues) substitutions were made. The peptide array was generated using the SPOT-synthesis technique on cellulose membranes [71,130,131,132]. This approach had been previously utilized to generate large libraries of antibacterial, immunomodulatory, and antibiofilm peptides [15,133,134]. The SPOT synthesized [131,132,135] peptides were tested for activity against *S. aureus* [71]. Peptides that contained one or more cation-π interactions did not show dramatically improved activity (1-2× MIC of parent aurein 2.2-Δ3), with the exception of two peptides, which had MICs that were 8× better and in which two cation-π interactions were present.

In order to understand why the modifications introduced into the two peptides mentioned above (namely, peptides 73 and 77, Table 2) resulted in more active analogues, a detailed study of the structure and mechanism of action was conducted [136]. Peptides 73 and 77 were found to be more bactericidal, less α-helical, and less likely to form clearly defined pores than aurein 2.2-Δ3. Indeed, the results obtained from DiSC_3_5 and pyranine assays only showed modest membrane depolarization and ion leakage caused by peptides 73 and 77. Overall, the data presented by Raheem et al. [136] suggested that the more active analogues behaved more like cell-penetrating peptides (CPPs), i.e., more like indolicidin [129,137,138,139], catestatin [140], penetratin [141], or other CPPs [142,143,144], than aurein 2.2-Δ3. Indeed, these cell-penetrating peptides have sequence features in common with peptide 73 and peptide 77. Moreover, catestatin is also α-helical for only part of the sequence [140]. As with other studies that involve AMP design, the higher antimicrobial activity that was associated with peptide 73 and peptide 77 was unfortunately also accompanied by higher hemolytic activity and cytotoxicity. This led us to explore bioconjugation and formulation strategies in order to mitigate these negative effects [4,60,71]. For an extensive review on these strategies, the reader is invited to consult [64] and references therein.

## 4. De Novo Designed β-Boomerang LPS Binding Antimicrobial and Antiendotoxic Peptides

A set of 12-residue long peptides were designed de novo for LPS binding and endotoxin neutralization. The primary structure of the first peptide in this series, YW12, is given in Table 3. This peptide was expected to assume a β-sheet or β-hairpin like structure and form a complex with LPS, thereby structurally mimicking β-sheet outer membrane proteins [145]. Analogues (Table 3) were designed, as follows: the four residue cationic motif KRKR at the centre of the sequence would interact with the negatively charged phosphate groups of lipid A of LPS and, hence, stabilize peptide anchoring onto the LPS surface; at the same time, aromatic and hydrophobic residues at the N- and C-termini of YW12 would insert into the multi-acyl chain domains of LPS. Due to their high intrinsic propensity for adopting β-sheet structure, aromatic residues Tyr and Phe and β-branched residues Ile and Val were introduced to further ensure that the desired structure would be adopted. An ensemble of 3-D structures of YW12 was determined while bound to LPS micelles using tr-NOESY NMR. YW12 did not adopt a β-sheet or β-hairpin structure, as per the design. However, the peptide interacted with LPS and folded into a unique structure resembling a ‘boomerang’ (Figure 3). Observed NOEs established proximity (≤5 Å) between residues Trp4 and Met9, which appeared to be critical in shaping the boomerang fold of the YW12 peptide, thereby resulting in an amphipathic structure. As most of the residues were in β-strand conformations, the structure was termed a β-boomerang. A molecular model of LPS and YW12 revealed multiple ionic and hydrogen bond interactions between the KRKR motif with the lipid A domain of LPS. The N- and C-termini residues made facile packing with the non-polar acyl chains of LPS. YW12 peptide showed selective binding to anionic detergents and lipids and assumed β-conformations in a membrane milieu. However, the YW12 peptide showed only weak activity in the neutralization of endotoxin and lacked antibacterial activity.

Based on the structure of YW12 (Figure 3), second generation of β-boomerang peptides were designed in order to enhance antimicrobial and antiendotoxic activity. Given that Trp4 and Met9 are essential for the stabilization of the β-boomerang fold [115], Met9 was substituted to aromatic amino acids, Trp, Phe, and Tyr, in order to enhance packing interactions by aromatic/aromatic ring stacking (Figure 3c). The analog peptides listed in Table 3 were examined for in vitro antiendotoxic and antibacterial activity. YI12WY peptide demonstrated bactericidal activity against Gram negative and Gram positive strains. YI12FF peptide did not display high antiendotoxic and antibacterial activity. Additionally, antiendotoxic and antibacterial activity of either YI12LL (Leu4/Leu9) and YI12AA (Ala4/Ala9) peptides could not be detected. Atomic-resolution structures of the active analogs YI12WW, YI12WF, YI12WY and the inactive analog YI12AA were determined by NMR in LPS micelles. The long range packing between residue Trp4 and the aromatic residue, Phe/Tyr/Trp, at the ninth position acted as an ‘aromatic lock’, which is critical for the activity and β-boomerang amphipathic structure of the peptides. The inactive analog also folded into a compact amphipathic structure, but it was rather open at the 4/9 end due to lack of the packing. The hexapeptide WKRKRF aromatic/cationic motif that potentially stabilizes the β-boomerang structure adopted a folded structure in LPS micelles, with mutual packing between Trp/Phe residues. This peptide sequence is the shortest segment known to bind LPS and was consequently termed as the ‘Structured LPS Binding Motif’ (see Section 2). The antiendotoxic and antibacterial activity of the β-boomerang peptides correlated well with the folded structure in LPS micelles. The active β-boomerang peptides were inserted into the micelles and disrupted higher order structural organization. As per the design, the four cationic residues at the centre of the β-boomerang appear to form ionic interactions with the phosphate groups of LPS and the hydrophobic residues at the N- and C-termini likely to destabilize the interchain packing of the acyl chains of LPS molecules.

N-terminal acylation and disulfide bonds can improve the potency and endotoxin neutralization activity of AMPs. The β-boomerang peptide YI12WF was further engineered to yield disulfide bonded lipopeptides for enhancing antibacterial and antiendotoxic activity. The analogs of YI12WF were designed by changing the primary structure slightly to yield YI13C (with a Cys residue at 11th position for dimerization through disulfide bond formation) and attaching either C4 (C4-YI13C) or C8 acylation (C8-YI13C) at the N-terminus. An extra Lys residue was incorporated to increase the solubility of the designed analogs [146]. In bacterial growth inhibition assays, C8-YI13C demonstrated lower MIC values against Gram negative and Gram positive bacteria as compared to C4-YI13C and YI13C. The disulfide bonded YI13C and both disulfide bonded lipopeptides were active in neutralizing endotoxin [146]. All three analogs were able to neutralize over 80% of endotoxin at the highest tested dose of 1 EU/mL at a concentration of 10 μM. Lipopeptides, C4-YI13C and C8-YI13C, displayed endotoxin neutralization at even lower concentration ranges. To examine the role of the two aromatic residues in activity, residues Trp4 and Phe10 in the potent peptide C8-YI13C were substituted with Ala, resulting in peptide C8-YI13CAA. This Ala analog was devoid of antibacterial and antiendotoxic activity. The disulfide bonded lipopeptides bind to LPS at appreciably higher affinity when compared to the parent YI12WF peptide. Based on ITC experiments, K_d_ was determined to be 0.23 μM and 0.45 μM for YI13C and C4-YI13C, respectively, whereas the parent peptide YI12WF binds to LPS with an estimated K_d_ of 4.5 μM. As a mode of antibacterial activity, the designed analogs neutralized the surface charge of bacteria and efficiently permeabilized the outer and inner membranes of bacterial cells. The lipopeptides were largely non-hemolytic and interacted with negatively charged lipids in model membranes, akin to the parent β-boomerang peptide [146]. Moreover, the binding of the lipopeptides and YI13C resulted in the fragmentation of LPS micelles into smaller sized aggregates, a phenomenon that has been correlated with antiendotoxin activity of LPS-interacting proteins and peptides. The atomic-resolution structure of C4-YI13C could be determined in free solution by NMR while using 207 NOE distant constrains. This is in sharp contrast to the parent peptide YI12WF and other β-boomerang peptides, which adopt random conformations in free solution. The 3-D structure clearly established the amphipathic β-boomerang structure of the two subunits of the disulfide bonded C4-YI13C peptide. The stabilization of the β-boomerang structure either by the disulfide bonded C4-YI13C in free solution or for the parent peptides in LPS micelles established a clear intrinsic tendency of the designed sequences to adopt such conformations. A third generation of β-boomerang peptides are currently in development. These peptides are cyclized at the N- and C-termini and they show even more potent antibacterial activity, including against drug resistant bacteria in in vitro and in vivo murine models. Wound healing activity was also observed for some of these short cyclized peptides (unpublished results).

## 5. Mining the Human Proteome for Novel AMPs

The human proteome could serve as a platform for the discovery of new AMPs, since such AMPs are likely to be non-toxic and non-immunogenic to humans. The human proteome contains bonafide host defence peptides and proteins, which protect against disease. Human AMPs, such as LL37, various defensins (α and β), histatins, and dermcidin, are expressed in skin, eyes, sweat, lungs, intestine lining, and the bladder [147,148,149]. These AMPs can directly kill pathogens or have immunomodulatory activity. Together with a number of proteins, such as lysozyme, different types of RNAse, chemokines, and psoriasin, AMPs are an integral part of human innate immunity. Aside from the bonafide AMPs and antimicrobial proteins, peptide fragments of native proteins can be active against invading pathogens [150,151,152]. These antimicrobial peptide fragments are usually hidden within the folded structure of proteins and they can only be released upon proteolytic digestion. For example, the antimicrobial fragments of blood coagulation protein thrombin were identified and characterized in terms of function and mode of action [152]. Thrombin derived peptide fragments are effective against a broad spectrum of bacteria in vitro and potentially also in in vivo wounds [153]. Thrombin peptides are active in modulating inflammation and are able to neutralize endotoxin, which plays an important role in sepsis [154]. In addition, several AMPs were discovered from DNA binding protein histones (H1, H2A, H2B, H3, H4, and H5) [155,156,157]. Histone-derived AMPs are rich in basic amino acids and adopt amphipathic, helical structures akin to many naturally occurring AMPs. Buforins, a group of potent amphibian AMPs, are postulated to be generated through pepsin-mediated proteolysis of histone 2A in the cytoplasm of gastric gland cells [31,158]. As a mode of bacterial cell killing, buforins bind to DNA, whereas other histone derived AMPs are thought to be membrane active [30,159]. Fragments of large antimicrobial proteins, such as lysozyme and RNases have been reported to exert bacterial growth inhibition. Several lysozyme derived peptide fragments, including a helix-loop-helix peptide, were characterized for antibacterial activity and membrane interactions [160,161]. Antibacterial peptides can be obtained from bacterial cell agglutinating eosinophil cationic protein (RNase 3) and potent analogs were further designed based on the structures [162,163]. An AMP termed lactoferricin derived from the N-terminal of iron binding protein lactoferrin and its shorter analogs were extensively characterized for antimicrobial activity and mechanism [164,165,166,167,168,169,170]. More recent studies have identified several fragments of ApoE protein with broad spectrum antimicrobial activity. A cryptic fragment of ApoE, residues 133–150 has been discovered using a novel bioinformatics method and was found to exert broad spectrum antibacterial activity, immunomodulation, and was less toxic to human cells [171]. Table 4 shows amino acid sequences of representative AMPs derived as fragments of proteins.

Protein zymogens often undergo proteolytic digestion during activation, releasing inhibitory prodomains into circulation. Isolated prodomains are bestowed with physiological functions and stability. They may also be potentially involved in host defence activity. Based on this premise, antimicrobial peptides were recently discovered from the prodomain of human furin, a serine protease. Proprotein Convertases (PCs) are a family of Ser proteases that are responsible for processing of numerous precursor proteins at the consensus cleavage sites [172,173]. Furin, which belongs to the PC family, is known to be widely expressed in various cells and functions in the constitutive secretory pathway [172]. Apart from endogenous processing of precursor proteins, several bacterial and viral proteins are processed and activated by furin. The 81-amino acid long prodomain of human furin undergoes auto-catalytic cleavage during protease activation. The prodomain of furin is highly cationic (pI ~ 12.2) and also rich in aromatic/hydrophobic residues. NMR structural studies of the prodomain revealed a compact molten globule state under physiological solution conditions and well-folded stable helical conformations in the membrane mimic trifluoroethanol [174,175]. No antimicrobial activity could be detected for full-length furin despite the fact that the physicochemical properties and helical secondary structure of the prodomain are commonly observed in a large proportion of AMPs. This was likely due to the fact that the potential antibacterial region(s) of the prodomain may be buried inside the core of the structure and are unable to exert cell killing activity. Consequently, a number of overlapping peptide fragments i.e., residues V67–R81, residues Q62–R81, residues R57–R81, residues S52–R81, and residues T47–R81, of increasing length were tested for antimicrobial activity from the cationic C-terminus of the prodomain [176]. However, none of the peptide fragments displayed detectable growth inhibitory activity against a panel of Gram negative and Gram positive bacteria. Two overlapping peptide fragments from the N-terminus of the prodomain, namely residues Q1-F35 (or QF35) and residues R24-V46 (or RV23) were also devoid of antibacterial activity, except against a strain of *E. coli*, where MICs of 20 μM and 4 μM for QF35 and RV23, respectively, were found. Interestingly, a central 26-residue peptide fragment or YR26 (Table 4) was highly active in inhibiting the growth of Gram negative and Gram positive bacteria, with MICs in the range of 2 to 4 μM. YR26 peptide was also able to kill drug resistant *S. aureus* and *S. epidermis* strains at low MIC values [176]. The antibacterial activity of several truncated variants, obtained by removing amino acids from the C-terminus of YR26, were further examined for the identification of potentially active shorter analogs. Antibacterial activity was largely retained for peptide fragments YR23 and YR20 (Figure 4). However, further truncations of residues resulted in significantly reduced bacterial cell killing, with the exception of the much shorter fragment YR12, which showed killing of most of the tested bacterial strains, with MICs somewhat lower than the other longer variants. YR12 may be further engineered to develop potent short AMPs. It is important to note that all tested peptides had amidated C-termini. All three active peptides, YR26, YR23, and YR20, are non-hemolytic and non-toxic to fibroblast cells. YR23 was further investigated for anti-inflammatory activity demonstrating inhibition of secretion of TNF-α and IL-β from LPS treated macrophage cells. Bacterial cell killing activity of the prodomain AMPs appears to occur via pore formation in membranes. Mechanistic investigations with live bacterial cells, LPS, and model liposomes revealed that the antibacterial peptides of the prodomain (a) disrupted the outer membrane-LPS permeability barrier and neutralized surface charge; (b) inserted into LPS micelles and destabilized the micellar structure; and, (c) interacted specifically with negatively charged lipids and caused the release of entrapped dyes from negatively charged POPC:POPG (3:1) liposomes. Interestingly, the binding of the prodomain peptides resulted fusion of liposomes (Figure 4).

The atomic resolution structure and backbone dynamics of YR26 peptide have been characterized while using standard methods and ^15^N relaxation experiments in negatively charged SDS micelles. YR26 adopted a helix-turn-helix structure, whereby a long N-terminal helix and a short C-terminal helix are connected by a tight turn (Figure 4). The helix-turn-helix structure of YR26 is rather non-amphipathic and it does not contain a clear disposition of distinct faces of cationic and hydrophobic residues. The striking feature of the helix-turn-helix structure is the presence of aromatic clusters at the N-terminal helix with a limited cationic patch, whereas the C-terminal helix accounted for most of the cationic surface. Based on the activity and a comparison of the structures of active and inactive truncated analogs, the membrane permeabilization, and bacterial cell killing afforded by YR26 peptide might be occurring due to cationic patches and aromatic cluster residues of the helix-turn-helix structure (Figure 4).

## 6. Conclusions

Arguably, at present, AMPs are vital candidates for fight against MDR and XDR bacteria. However, rational design, engineering and chemical modifications are necessary for the conversion of AMPs templates into in vivo active stable antibiotics. In this review, we have discussed the short helical AMPs temporins and aurein of frog skin, as well as de novo designed β-boomerang AMPs. Many analogs of temporins and aurein displayed improved antibacterial potency and higher therapeutic index. In addition, we highlight the discovery of novel AMPs that are derived from fragments of human proteins. The on-going efforts across the globe involving basic research and clinical trials of AMPs may deliver much needed next generation antibiotics.

## Figures and Tables

**Figure 1 ijms-21-05773-f001:**
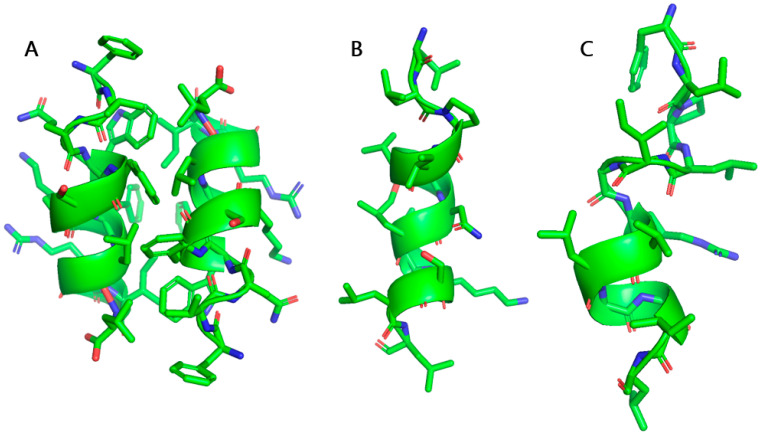
Structures of (**A**) the TL dimer; (**B**) TA; and, (**C**) TB in LPS. TL assumes an anti-parallel dimeric structure, stabilized by packing of aromatic and hydrophobic residues. The monomeric helical structures of TB and TA were obtained in the presence of TL peptide. The helical fold is represented by a ribbon. The figure was prepared using PyMOL molecular modelling software. The atomic coordinates of TL and TB were deposited in the BMRB data bank with accession numbers 21008 and 21005, respectively. The atomic coordinates of TA can be found in the Protein Data Bank with accession number 2MAA.

**Figure 2 ijms-21-05773-f002:**
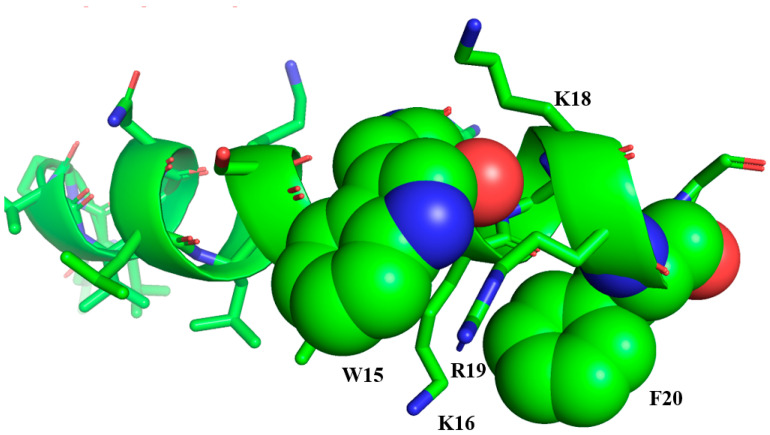
Structure of LG21 in micelles: the TB segment forms a well-defined α-helical structure, while the β-boomerang motif packs with the helix to form a lollypop. The atomic coordinates of LG21 are available upon request from the authors.

**Figure 3 ijms-21-05773-f003:**
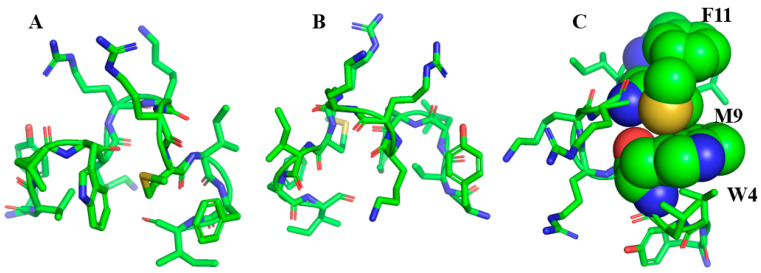
Structure of YW12 in LPS (pdb accession code: 2O0S): the β-boomerang shape inspired the design of the other peptides listed in Table 3. Sidechain dispositions of the β-boomerang structure are shown in two different orientations (**A**,**B**). The designed peptide assumed an amphipathic structure, with the central cationic residues K^5^RKR^8^ exposed to the outer periphery, whereas the N- and C-termini aromatic and non-polar residues sustained inward packing interactions. (**C**) Sidechains of residues W4, M9, and F11 of YW12 are in close packing interactions that are found to be critical for the activity.

**Figure 4 ijms-21-05773-f004:**
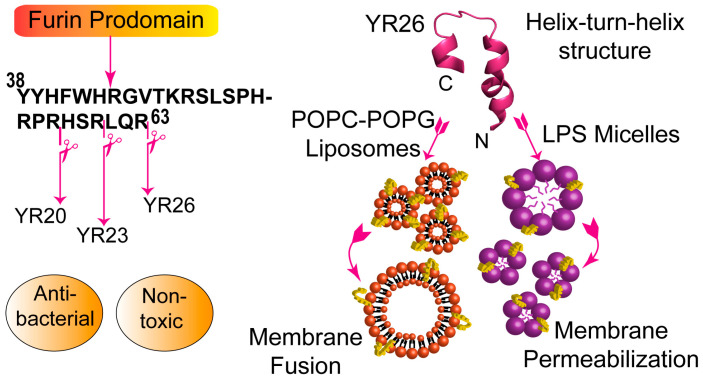
Truncation of furin prodomain to yield antimicrobial YR26, YR23, and YR20 (left). Structure and interaction of YR26 with model membranes (right). YR26 peptide assumed helix-turn-helix structure in SDS micelle and caused liposome fusion and LPS micelle dissociation. The figure was adapted from reference 176. The pdb accession code of YR26 is 6A8Y.

**Table 1 ijms-21-05773-t001:** Designed antimicrobial peptides (AMPs) derived from the frog skin Temporins temporin-1Ta (TA), temporin-1Tb (TB), and temporin-1Tl (TL).

Name	Sequence	Net Charge
Temporin A (TA)	FLPLIGRVLSGIL-NH_2_	+2
Temporin B (TB)	LLPIVGNLLKSLL-NH_2_	+2
Temporin L (TL)	FVQWFSKFLGRIL-NH_2_	+3
TB_KKG6A	KKLLPIVANLLKSLL-NH_2_	+4
TB_L1FK	FLPIVGLLKSLLK-NH_2_	+3
TB-YK	KKYLLPIVGNLLKSLL-NH_2_	+3
TL (P3, D-P10)	FVPWFSKFLpRIL-NH_2_	+2
TL analog 9	FVPWFSKFlkRIL-NH_2_	+4
TL analog 10	FVPWFSKFlWRIL-NH_2_	+3
TA-β-boomerang (FG21)	FLPLIGRVLSGILGWKRKRFG-NH_2_	+6
TB-β-boomerang (LG21)	LLPIVGNLLKSLLGWKRKRFG-NH_2_	+6

**Table 2 ijms-21-05773-t002:** Designed AMPs derived from the frog skin aurein 2.2.

Name	Sequence	Net Charge
aurein 2.2	GLFDIVKKVVGALGSL-NH_2_	+2
aurein 2.3	GLFDIVKKVVGAIGSL-NH_2_	+2
aurein 2.2-Δ3	GLFDIVKKVVGAL-NH_2_	+2
peptide 73	RLWDIVRRWVGWL-NH_2_	+3
peptide 77	RLWDIVRRVWGWL-NH_2_	+3

**Table 3 ijms-21-05773-t003:** De Novo Designed β-boomerang AMPs.

Name	Sequence	Net Charge
YW12	YVLWKRKRMIFI-OH	+4
YI12WF ^1^	YVLWKRKRFIFI-NH_2_	+5
YI12WY	YVLWKRKRYIFI-NH_2_	+5
YI12WW	YVLWKRKRWIFI-NH_2_	+5
YI12FF	YVLFKRKRFIFI-NH_2_	+5
C4/C8-YI13C	C4/C8-YVLWKRKRKFCFI-NH_2_	+6

^1^ Although the parent peptide was called YW12, the designed analog peptides were referred using their first and last amino acid residues. Also, amino acid residues replaced at positions 4 and 9 are included for ease of identification.

**Table 4 ijms-21-05773-t004:** Fragments of Intra-cellular Proteins as Potent AMPs.

Name	Sequence	Net Charge
TCP (C-terminal of thrombin)	GKYGFYTHVFRLKKWIQKVIDQFGE	+5
Buforin II	TRSSRAGLQFPVGRVHRLLRK	+8
Human lysozyme	DNIADAVACAKRVVRDPQGIRAWVAWRNR	+4
Lactoferricin	KCFQWQRNMRKVRGPPVSCIKRDS	+6
ApoE	LRVRLASHLRKLRKRLLR	+10
Prodomain of Furin (YR26)	YYHFWHRGVTKRSLSPHRPRHSRLQR	+12

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
