# Peer review of "Design, Engineering and Discovery of Novel α-Helical and β-Boomerang Antimicrobial Peptides against Drug Resistant Bacteria"

_ijms, 2020, doi:10.3390/ijms21165773_

Round 1
Reviewer 1 Report
This review manuscript about novel antimicrobial peptides was well written by Dr. Bhattacharjya and Dr. Straus. It is absolutely suitable for IJMS.
Minor concerns
Ref. 36 "Science (80-.)." to "Science".
Ref. 77 "Luisa Mangoni M." to "Mangoni, M. L"
Author Response
Dear Professor Galzitskaya,
Please find attached a revised manuscript entitled “Design, Engineering and Discovery of
Novel α-helical and β-boomerang Antimicrobial Peptides Against Drug Resistant Bacteria” by
Surajit Bhattacharjya and myself. We have addressed the reviewers’ comments by tracking the
changes in the manuscript and detail the changes below as well.
Reviewer 1: This review manuscript about novel antimicrobial peptides was well written by Dr. Bhattacharjya and Dr. Straus. It is absolutely suitable for IJMS.
We thank Reviewer 1 for their positive comments.
Minor concerns
Ref. 36 "Science (80-.)." to "Science".
This has been corrected.
Ref. 77 "Luisa Mangoni M." to "Mangoni, M. L"
This has also been corrected.
Reviewer 2 Report
The current review by Bhattacharjya and Straus is interesting and very complete. The authors support their statement with appropriate references and the text is well illustrated. I have only few minor points that I believe will be able to improve this manuscript. Otherwise I believe that this story will be of general interest for your readership.
Questions:
Are bacteria resistant to colistin (or any other antibiotics targeting LPS) sensitive to AMP? This might be an interesting point to discuss in the introduction.
Can hypoacetylated LPS binds to boomerang AMP?
Are human-derived AMP more stable for drug-development purposes?
Minor points:
It would be helpful if the authors refer to the PDB reference number for each structure displayed in each figure legends.
It could be nice to include a quantification (++, +, -) of the activity if each peptide in their respective tables. It would be easier for the reader to compare them.
MIC definition was first mentioned at page 6. However, he was first mentioned (but not defined) at page 4. MIC should be defined the first time it is mentioned in the text.
Line 443/444: The ‘’not’’ on line 444 should be removed.
Please let me know if you have any questions,
Kind regards
Dave
Author Response
Reviewer 2: The current review by Bhattacharjya and Straus is interesting and very complete. The authors support their statement with appropriate references and the text is well illustrated. I have only few minor points that I believe will be able to improve this manuscript. Otherwise I believe that this story will be of general interest for your readership.
We thank Reviewer 2 for their positive comments.
Questions:
Are bacteria resistant to colistin (or any other antibiotics targeting LPS) sensitive to AMP? This might be an interesting point to discuss in the introduction.
Yes, we agree. We have added the following sentence and reference on lines 63-65: “For example, Gram negative bacteria which are colistin resistant can be sensitive to AMPs: LL-37, cecropin A, and magainin 1 demonstrated lethality against colistin resistant K. pneumoniae (Hashemi MM, Rovig J, Weber S, Hilton B, Forouzan MM, Savage PB. Susceptibility of Colistin-Resistant, Gram-Negative Bacteria to Antimicrobial Peptides and Ceragenins. Antimicrob Agents Chemother. 2017;61(8):e00292-17).” Note this means that we
have added a new reference to the reference list and the formatting is changed in this section.
Can hypoacetylated LPS binds to boomerang AMP?
As we have not examined such interactions, we would not like to comment.
Are human-derived AMP more stable for drug-development purposes?
As outlined in the text, unmodified AMPs, regardless of the source, are generally
sensitive to degradation by proteases. Thus this includes AMPs derived from human proteins.
However, human AMPs are likely to be less immunogenic compared to AMPs derived from other sources. We had stated this on lines 400-401 and believe it should be clear.
Minor points:
It would be helpful if the authors refer to the PDB reference number for each structure displayed in each figure legends.
The figure legends have been amended to include the PDB or BMRB codes.
It could be nice to include a quantification (++, +, -) of the activity if each peptide in their respective tables. It would be easier for the reader to compare them.
Unfortunately, this would be difficult to do, as the activity is measured against a number of different strains. We suggest the reader goes to the original texts to get a complete picture of activity.
MIC definition was first mentioned at page 6. However, he was first mentioned (but not defined) at page 4. MIC should be defined the first time it is mentioned in the text.
We apologize for this. The definition has been moved to page 4 and removed from page 6.
Line 443/444: The ‘’not’’ on line 444 should be removed.
This has been done.
We thank the reviewers for their insightful comments, which have helped to improve the manuscript. We hope that we have addressed all concerns in a satisfactory manner. If you require information, please do not hesitate to contact me. We look forward to hearing back from you soon.
Sincerely